# Knowledge Enhanced Image Captioning for Fashion Products

## Abstract

The field of image captioning has witnessed a surge in attention, particularly in the context of e-commerce, where the exploration of automated fashion description has gained significant momentum. This growing interest can be attributed to the increasing influence of visual language and its impact on effective communication within the fashion industry. However, generating detailed and accurate natural language descriptions for fashion items remains a topic of intense discussion. This paper introduces an innovative approach that specifically addresses this challenge. Our approach integrates a knowledge base into the widely adopted end-to-end architecture, thereby enhancing the availability of comprehensive data about fashion items. We design a mode mapping network that facilitates the fusion of attribute features extracted from the knowledge base with image features. Additionally, we introduce a filter strategy to enhance the quality of the generated descriptions by selecting the best result among the candidate sentences generated through beam search using a language model. Through extensive experimentation and evaluation, our proposed method demonstrates superior performance in the task of fashion description, surpassing the performance of state-of-the-art approaches in this domain.

## 1 Introduction

The task of describing fashion items in the e-commerce domain differs from traditional image captioning tasks, as highlighted in previous works (Xu et al., 2015; Anderson et al., 2018; Huang et al., 2019), which primarily focus on generating descriptive sentences from a given image. This difference stems from the unique requirements of fashion product descriptions. Not only should these descriptions accurately reflect the product's intricate details and features, but also they should be vivid and illustrative, effectively capturing users' attention while showcasing the distinctive characteristics of the item.

With the rise of fashion networking platforms, there is a noticeable trend towards automating the product description process, especially when dealing with a large volume of images (Hacheme & Sayouti, 2021). However, existing models that are primarily designed for natural images have limitations when it comes to generating diverse and non-repetitive descriptions (Vinyals et al., 2015; Xu et al., 2015). These models often produce descriptions that follow a templated structure and lack the desired level of diversity and comprehensive details. Consequently, effectively highlighting the key elements and attributes depicted in the image becomes challenging.

To address these challenges, we present a novel approach named KEIC (i.e. Knowledge Enhanced Image Captioning). Our approach aims to capture and represent the fine-grained features of products depicted in images, while also emphasizing key information during the caption generation process. We integrate a fine-grained knowledge base for products into the conventional end-to-end architecture. This enhancement enriches the image encoding process and enables a more profound semantic analysis of the images. The method enhances the overall representation of the images, leading to more accurate and contextually relevant descriptions of the fashion products. We incorporate the fused feature as a prefix to facilitate the generation of image descriptions. By leveraging this approach, it is anticipated that the limitations of existing models, such as the lack of diversity and comprehensive details, can be overcome, resulting in more accurate and compellingly product descriptions in the e-commerce context.

Our main contributions can be summarized as follows:

1. We built a knowledge base for fashion items and incorporate it into the end-to-end architecture, which provides access to a wealth of multidimensional data about fashion items.

2. We design a new mode mapping network that facilitates the fusion of attribute features extracted from the knowledge base with image features. It allows for a more comprehensive representation of fashion items, capturing both visual and attribute information.

3. To improve the quality of the generated descriptions, we introduce a filter strategy. This strategy involves selecting the best result from the candidate sentences generated through beam search.

4. In order to evaluate the effectiveness of our proposed method, we conducted comprehensive experiments, including the comparison with the state-of-the-art approaches, ablation study, and case study analysis. Experimental results confirm the effectiveness of our method.

## 2 RELATED WORK

In recent years, significant research efforts have been dedicated to the field of image captioning (Hossain et al., 2019). Initially, a common approach involved generating simple template sentences that were then populated with the outputs of object detectors or attribute predictors (Farhadi et al., 2010). However, with the emergence of deep learning, captioning methods started adopting recurrent neural networks as language models, leveraging the outputs of convolutional neural networks (CNN) to encode and guide the generation of captions (Kiros et al., 2014). Building upon this foundation, Xu et al. (Xu et al., 2015) introduced an attention mechanism that operates on the spatial output grid of the convolutional layer. This mechanism enabled the model to influence the generation process by selectively attending to specific elements in the grid while generating each word. By dynamically attending to different regions of the image, the model could generate more contextually relevant and detailed descriptions.

The method of Anderson et al. (Anderson et al., 2018) combines bottom-up Faster R-CNN object detection and top-down weighted region prediction, which enables the model to generate dense detection sets and enhance feature representations through auxiliary training on the visual genome dataset. The reticulation decoder designed by Cornia et al. (Cornia et al., 2020) contains a reticulation operator that independently regulates the contributions of all coding layers and a gate mechanism that weights these contributions under the guidance of textual queries to exploit the information of all coding layers. They consider all the encoding layer information in order to better meet the requirements of text queries.

Huang et al. (Huang et al., 2019) proposed an extended attention operator to refine visual features by weighting the final attention information through a context-guided gate mechanism, where the output of self-attention is multiplied with the query join computing information and the gate vector. In recent years, e-commerce has attracted more and more attention, and many scholars have launched attempts in the field of e-commerce to provide different solutions. Han et al. (Han et al., 2022) proposed fashion-centric visual and language (V+L) representation learning framework FashionViL, but there are still challenges in fine-grained cross-modal alignment.

Zhu et al. (Zhu et al., 2021) proposed a new method, K3M, which designed a structural aggregation module to integrate information from image, text, and knowledge modalities. Moratelli et al. (Moratelli et al., 2023) proposed grid relational Self-attention (GSA) and Gated Augmented decoder (GED) to strengthen visual representation and dynamically measure the contribution of different views to the target word and applied it to a Transformer model for the fashion item captions task. However, these methods mainly focus on entities in text modality and are still insufficient for cross-modal interaction.

The object detection approach proposed by Li et al. (Li et al., 2020) focuses on identifying specific objects within an image, which is useful for general image understanding but still does not address the specific requirements of e-commerce tasks. Similarly, scene graph parsing, as introduced by Cui et al. (Cui et al., 2021), aims to extract structured representations of objects and relationships in an image. While these techniques consider fine-grained information has shown promise in various domains, their effectiveness in e-commerce tasks appears to be limited.

Considering the recent advancements in multimodal large-scale models for visual language understanding, a prominent approach entails leveraging pre-trained models and fine-tuning them using e-commerce data. This approach has demonstrated promising results in various studies (Mirchandani et al., 2022; Zhuge et al., 2021; Wang et al., 2022).

Clipcap is an innovative approach that shows great potential in the field of image captioning. It capitalizes on the utilization of pre-trained models CLIP (Contrastive Language-Image Pre-Training) (Radford et al., 2021) and introduces a streamlined architecture to generate captions for images. The key idea behind this approach is to leverage the existing semantic understanding encoded in pre-trained models and adapt it to the style of the target dataset, rather than learning entirely new semantic entities (Mokady et al., 2021). Although Clipcap demonstrates great effectiveness in image captioning tasks, its performance in fine-grained image tasks is comparatively subpar in practical applications (Zhong et al., 2022).

In the context of e-commerce, the challenges go beyond mere object detection and scene understanding. The goal is to generate comprehensive and accurate descriptions that capture the fine-grained details and relevant attributes of the products. Therefore, addressing the unique characteristics of e-commerce data, such as product-centric information and the importance of relevant attributes, are essential for achieving satisfactory results in e-commerce image captioning tasks (Wang et al., 2024).

## 3 METHODOLOGY

The primary objective of image captioning is to develop a model capable of generating meaningful captions for unseen input product images. This problem revolves around two modalities: image and text. In this paper, we utilize the symbol $I$ to represent an image and $C$ to represent the corresponding image caption, which is the textual description associated with the image. To train the model, we employ a dataset consisting of paired images and captions denoted as $\{I^i, C^i\}_{i=1}^n$. Here, $n$ signifies the total number of image-caption pairs in the dataset, and the caption $C^i$ can be represented as a sequence of tokens $c_1^i, c_2^i \ldots c_q^i$, in which $q = |C^i|$.

As depicted in Figure 1, the framework is divided into three modules. Feature Encoder: The fine-grained commodity knowledge base is introduced into the feature encoding stage to enrich the image features. Mode Mapping Network: The attribute features and image features are integrated so that the acquired features contain more elements of the product. Inference and Generation: It handles downstream tasks to infer and generate the product descriptions.

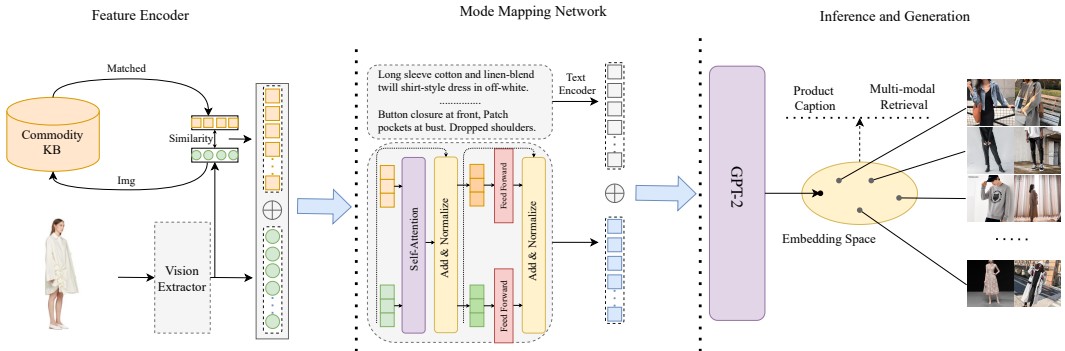

Figure 1: Framework of the knowledge enhanced image captioning. It consists of three components: Feature Encoder, Mode Mapping Network, Inference and Generation.

### 3.1 FEATURE ENCODER THAT ENRICH IMAGE FEATURES

In the feature encoder module, our objective is to capture rich, diverse, and fine-grained feature information of fashion images. To accomplish this, we enhance the representation of fashion image features by incorporating product attribute tags to capture a more comprehensive and detailed un-

derstanding of the fashion items in the image. We propose a diverse product attribute knowledge base, which allows us to acquire multidimensional product feature information.

Given an input image $I$, we utilize CLIP's visual encoder (Dosovitskiy et al., 2021) to acquire detailed feature information from product images. This visual encoder processes the image $I$ and produces corresponding image features denoted as $F_I$. These image features are represented as a $d$-dimensional feature vector, capturing the essential characteristics of the input image $I$ as shown in Equation 1.

$$F_I = encode\_image(I) \tag{1}$$

To enhance the understanding of the knowledge within the image, we extract product attribute tags from the image. The knowledge base serves as a repository for product category directories, noun chunks, and product attributes. The construction of this knowledge base is achieved by leveraging the textual information present in the captions associated with the images. The computation of the text feature $F_t$ is shown in Equation 2.

$$F_t = enocde\_text(t) \tag{2}$$

The knowledge base **D** is composed of attribute tags and their corresponding textual features, forming a collection of key-value mapping pairs $F_t \rightarrow t$. Here, $F_t$ represents the text feature of the attribute tag $t$. Given an input image, we leverage the image feature extraction process described earlier to obtain the image features $F_I$. We then compare these image features with the text features in the knowledge base to identify relevant attribute tags. To extract the relevant attribute tags, we employ a similarity measure to compute the similarity scores between the image features $F_I$ and the text features $F_t$ of the attribute tags in **D**. The attribute tags with the highest similarity scores are considered the most relevant to the depicted products.

To ensure the relevance and diversity of product descriptions, we have developed a tag extraction algorithm that considers both aspects. The relevance of an attribute tag is determined by computing the similarity between the image features $F_I$ and the text feature $F_t$ associated with the attribute tag. This similarity calculation is represented by Equation 3.

$$rel(I,t) \stackrel{def}{=} Sim(F_I, F_t) \tag{3}$$

Here, $Sim$ refers to the similarity measure used to quantify the similarity between the image features $F_I$ and the text feature $F_t$. This measure provides a numerical value indicating the degree of similarity between the two features. Cosine similarity is the most common similarity measure function.

The diversity is obtained from the dissimilarity between the text feature $F_t$ and the attribute tag $s$ in the current tag set $S$ as depicted in Equation 4.

$$div(t,S) \stackrel{def}{=} 1 - \max_{s \in S} Sim(F_t, F_s)] \tag{4}$$

The final score assigned to an attribute tag is a balance between its relevance and diversity as depicted in Equation 5.

$$tag\_score(t, I, S) = \lambda \cdot rel(I,t) + (1 - \lambda) \cdot div(t,S) \tag{5}$$

in which, $\lambda$ is the weight coefficient to adjust the relevance and diversity of searching result.

The target of knowledge enhanced attribute tags extraction is to find attribute tag who have the highest scores as depicted in Equation 6

$$tag\_extract(T, I, S) = \arg \max_{t \in T \setminus S} tag\_score(t, I, S) \tag{6}$$

We perform a KNN search in the tags in **D** to select the top $k$ attribute tags as the attribute tags of the product image $I$. By mining and extracting these relevant attribute tags, we obtain a comprehensive understanding of the product attributes present in the image. These attribute tags serve as valuable descriptors that contribute to generating accurate and informative fashion descriptions.

## 3.2 Mode Mapping Network for Multimodal Feature Interaction

To ensure sufficient interaction between multidimensional features and ensure that the image feature and corresponding attribute tag features can be perfectly aligned and smoothly inputted into the language model, we introduce a spatial mapping method to complete the fusion of multidimensional product features and modality transformation.

After querying each image in the knowledge base, a set of corresponding tags is obtained. To process these tags, we first encode the entire group of tags into a set of features denoted as $F_S$, as illustrated in Equation 7.

$$F_S = encode\_text(t_1, t_2, \ldots, t_{|S|}), \text{where } t_i \in S \tag{7}$$

To ensure consistency between the fused image features and the spatial dimension of the language model in the subsequent stage, a spatial network mapping is performed. This transformation is achieved using a multi-layer perceptron (MLP). The feature matrix of the image features is converted to $V$ as depicted in Equation 8, where $V \in \mathbb{R}^{l \times m}$, $l$ represents the length of the image feature sequence and $m$ represents the dimension of the mapping network.

$$V = MLP(Concat(F_I, F_S)) \tag{8}$$

The concatenated embedding matrix is then processed through a mapping network, utilizing a multilayer attention mechanism to learn common features. The output of this mapping network is denoted as $E \in \mathbb{R}^{l \times m}$, as shown in Equation 9.

$$E = ModeMap(Concat(V, C)) \tag{9}$$

Here, ModeMap represents the mapping network, and $V$ is the concatenated embedding matrix. The learning vector $C \in \mathbb{R}^{l \times m}$ is initialized randomly and has the same dimensions as the image feature matrix $V$. Its objective is to encapsulate the shared characteristics between the original image features and the entity attribute features sourced from the knowledge base.

Through the multilayer attention mechanism, the mapping network learns to extract and emphasize relevant information from the concatenated embedding matrix. This enables the network to identify and capture the shared characteristics and correlations between the original image features and the attribute features.

## 3.3 Inference and Generation Based on Language Model

In this study, we utilize the GPT-2 language model as the decoder and employ the extracted visual features as pre-defined prompt words to generate relevant descriptions. Given a product image $I$, the caption is a sequence of $q$ tokens $C = \{c_1, c_2, \ldots, c_q\}$. We first calculate the embedding matrix by feature encoding $F_C \in \mathbb{R}^{h \times m}$, in which $m$ is the embedding dimension of each token, $h$ is the length of the longest caption. With the embedding matrix $E$ mapped in the encoding stage, the input of the language model is obtained through concatenating the two tensor $H = Concat(E, F_C)$, in which $H \in \mathbb{R}^{(l+h) \times m}$. In the training process, the content of the image continuously affects the generation of the model and reduces the exposure bias in the generation process.

The attention scores are then used to weight the values, resulting in a weighted sum that represents the output of the attention head. This weighted sum captures the contextual dependencies and relationships between different positions in the input, allowing the model to focus on the most relevant information for each position as shown in Equation 10.

$$Att(q, i) = \left[ softmax(\frac{Q_{(q,i)} \times K^T_{(q,i)}}{\sqrt{d_k}}) \odot M_{(q,i)} \right] V_{(q,i)} \tag{10}$$

Here, $M_{(q,i)}$ represents a masking matrix. The input is transformed using $W_{Q_{(q,i)}}$, $W_{K_{(q,i)}}$, and $W_{V_{(q,i)}}$ to obtain query, key, and value representations.

By incorporating multiple attention heads, each with its own set of learnable parameters, the model can capture different types of relationships and dependencies within the input. After weighted summation across multiple attention heads, we acquire processed feature information. Using the weight matrix $W_{O_q}$, we map the concatenated self-attention to the final output $Z_q$. To facilitate input into the FFN layer, the vector undergoes a linear mapping $W_{O_q}$ as depicted in Equation 11.

$$Z_q = Concat(Att(q, 1), Att(q, 2), \dots Att(q, h)) \cdot W_{O_q} \tag{11}$$

in which, $Concat(\cdot)$ represents the concatenation operation, and $W_{O_q}$ is a weight matrix.

In the model's output, when the last decoder layer produces an output vector, the model multiplies this vector by a huge embedding matrix $W_g$ to compute the relevance scores between this vector and all word embedding vectors in the vocabulary. We choose the token with the highest score as the output token. After selecting the token feature matrix $H_q$, the model learns to predict the next token until all words have been generated or until a token representing the end of the sentence is output. The computation is shown in Equation 12.

$$P(x_{t+1} \mid x_1, x_2 \dots x_t) = softmax(H_q \cdot W_g) \tag{12}$$

Here, $W_g \in \mathbb{R}^{d \times V}$ is the embedding matrix, where each row corresponds to a token in the model's vocabulary. The softmax operation calculates the probability of each token in the vocabulary.

To enhance the quality of the generated content, we propose a filtering strategy. Specifically, we employ beam search to generate multiple candidate sentences simultaneously. The process is relaxed to select the top-$k$ tokens. Subsequently, we leverage the text encoder from CLIP to compute features and calculate the similarity between the generated sentences and the corresponding image features. By ranking the sentences based on the computed scores, we select the sentence that exhibits the highest similarity with the image features as the final generated description. This step helps to ensure that the generated content is more aligned with the visual information depicted in the image.

## 4 EXPERIMENTS

### 4.1 EXPERIMENTAL SETTINGS

#### 4.1.1 DATASETS

In this study, we conducted an evaluation of various models using the Fashion-Gen dataset, which comprises a collection of $293,008$ images. Each image in the dataset corresponds to a fashion item. Moreover, the dataset includes expert-written paragraph descriptions that provide detailed information about the fashion items.

To ensure consistency and enable unbiased comparisons with other models, we applied a standardized partitioning strategy to split the dataset. Specifically, for training purposes, we utilized $260,480$ images, while reserving a set of $32,528$ images exclusively for testing our models.

#### 4.1.2 METRICS

To evaluate the quality of fashion captions generated by our method and other comparative approaches, we employed six widely used evaluation metrics: BLEU-1, BLEU-2, BLEU-3 , and BLEU-4 (Papineni et al., 2002), METEOR (Denkowski & Lavie, 2014), and CIDEr-D (Vedantam et al., 2015). These metrics provide quantitative measures to assess the similarity and quality of the generated sentences compared to the ground truth sentences. Higher values of these evaluation

metrics indicate that the generated sentences are not only closer to the reference sentences but also of higher quality in terms of linguistic coherence, fluency, and relevance to the fashion items.

### 4.1.3 EXPERIMENTAL PARAMETERS

When we build the product attribute knowledge base, we extracted $14,920$ noun phrases from the Fashion-Gen dataset as attribute tags. We set $k$ to 5 to select the top-5 attribute tags. To extract the attribute tags, we use cosine similarity as the similar measure to rank the attribute tags in the knowledge base, and we set $\lambda = 0.5$ to optimal balance diversity and relevance.

The remaining configuration parameters were defined as follows. The temperature parameter, which controls the randomness of the generated text, was set to 1. A higher temperature value allows for more diverse and exploratory outputs. For the top-$k$ filtering technique, the parameter $k$ was set to 10. Similarly, for the top-$p$ filtering technique, the parameter $p$ was set to $0.7$. During the training phase, the number of epochs was set to $80$. The batch size was set to $40$. We set the prefix length $l = 10$ for the concatenated embedded matrix learned from the mapping network. For optimization, we use AdamW (Kingma & Ba, 2015) with learning rate 0.00002 and 5000 steps to warm up.

All comparisons are conducted under the same experimental environment, and all training and inference tasks are completed on a single NVIDIA A4000 server GPU to evaluate the performance of the models.

### 4.2 EXPERIMENTAL RESULT

In this study, we conducted a thorough performance comparison between our proposed model and other state-of-the-art models in the field of fashion captioning. Our evaluation encompassed a comprehensive analysis of various metrics at different levels, allowing for a detailed assessment of the model's performance. Furthermore, we performed ablation experiments on our proposed model to validate the rationale behind each component. In addition to quantitative evaluations, we also conducted a case study to qualitatively assess the generated fashion captions.

### 4.2.1 OVERALL EVALUATION

In our evaluation on the Fashion-Gen dataset, we compared the performance of our method with five popular existing methods. These methods were categorized based on their decoder architecture into two categories:

1. LSTM-based Methods:
    (a) Att2In (Xu et al., 2015): This method represents a classic baseline that utilizes LSTM as the decoder for generating fashion captions.
    (b) UpDown (Anderson et al., 2018): Another LSTM-based baseline model that incorporates bottom-up and top-down attention mechanisms to generate fashion captions.
    (c) AOA (Huang et al., 2019) A third LSTM-based baseline method that employs an attention-over-attention mechanism to enhance the quality of generated fashion captions.

2. Transformer-based Methods:
    (a) Mesh-Memory (Cornia et al., 2020): This model is built upon the Transformer architecture and serves as a classic baseline for fashion caption generation.
    (b) Clipcap (Mokady et al., 2021): This method utilizes the GPT-2 language model as the decoder, making it the baseline model for incorporating GPT-2 in fashion captioning. It also serves as the baseline model for our proposed method.

The experimental results conducted on the Fashion-Gen dataset are presented in Table 1. The best and second-best results in our experimental findings are highlighted using the typographical emphasis of bold and underlined text. As per the evaluations, our proposed method demonstrates superior performance compared to the other models across the majority of the evaluation metrics. Notably, our method achieves remarkable scores in terms of CIDEr-D metric, surpassing the Mesh-Memory (Cornia et al., 2020) and ClipCap (Mokady et al., 2021) models by $13.4\%$ and $3.2\%$ respectively.

Table 1: Performance Comparison on Fashion-Gen Dataset[%]

| Model | BLEU-1 | BLEU-2 | BLEU-3 | BLEU-4 | METEOR | CIDEr-D |
|---|---|---|---|---|---|---|
| Att2In | 48.4 | 37.7 | 28.4 | 21.7 | 26.0 | 100.3 |
| UpDown | 48.6 | 38.2 | 28.9 | 22.1 | 26.5 | 101.1 |
| AOA | 50.6 | 39.7 | 30.3 | 23.1 | 28.3 | 107.2 |
| Mesh-Memory | 51.9 | 40.5 | 30.6 | 23.2 | 27.7 | 107.7 |
| GSA-GED | **53.2** | **41.3** | 31.0 | 23.5 | 27.7 | 112.1 |
| ClipCap | 51.1 | 40.0 | _31.4_ | _25.5_ | _29.3_ | _117.9_ |
| KEIC(Ours) | _52.1_ | _40.9_ | **32.3** | **26.2** | **30.0** | **121.1** |

These results highlight the effectiveness of our method in generating high-quality fashion captions, particularly in terms of capturing the diversity and relevance of the generated captions to the reference annotations. It is worth noting that the metrics of our KEIC model outperform all the metrics of the baseline model ClipCap, further emphasizing the benefits of incorporating a knowledge base during the encoding stage and employing optimization strategies during the inference stage.

These findings validate the efficacy of our proposed approach in leveraging knowledge and optimizing the caption generation process, leading to improved performance and more accurate and contextually meaningful fashion captions on the Fashion-Gen dataset.

### 4.3 ABLATION STUDY

Our proposed architecture incorporates different components to demonstrate the enhanced performance of the model. One of the components is the knowledge-enhanced attribute extraction module (KB), while the other is the filtering strategy (FS).

In Table 2, we present the ablation study results, which reveal the positive impact of each component on the overall model performance. We establish a baseline in the first row, followed by the inclusion of the knowledge base (KB) in the second row and the filtering strategy (FS) in the third row.

Table 2: Ablation study result.

| Base | KB | FS | BLEU-1 | BLEU-2 | BLEU-3 | BLEU-4 | METEOR | CIDEr-D |
|---|---|---|---|---|---|---|---|---|
| ✓ | | | 50.5 | 39.5 | 31.1 | 25.2 | 29.1 | 114.8 |
| ✓ | | ✓ | 51.3 | 40.2 | 31.7 | 25.7 | 29.5 | _119.2_ |
| ✓ | ✓ | | _51.8_ | _40.8_ | _32.0_ | **26.2** | _29.8_ | _119.2_ |
| ✓ | ✓ | ✓ | **52.1** | **40.9** | **32.3** | **26.2** | **30.0** | **121.1** |

Both the knowledge base and the filtering strategy demonstrate improvements across all evaluation metrics when compared to the baseline. Notably, the knowledge base component shows a significant increase of $4.4\%$ in the CIDEr-D metric. When both components are integrated, they exhibit a combined improvement of $6.3\%$ over the baseline across all metrics.

We observe that the introduction of the knowledge base has the most substantial impact on the baseline model. We attribute this improvement to the extensive incorporation of nouns and entities from the knowledge base, which strengthens the extraction of object features and leads to more accurate descriptions.

These findings highlight the effectiveness of both the knowledge base and the filtering strategy in enhancing the model's performance. The integration of the knowledge base provides valuable information for attribute extraction, while the filtering strategy further refines the generated content. The combination of these components yields notable improvements in generating more accurate and contextually meaningful descriptions.

### 4.4 CASE STUDY

In addition to quantitative evaluation, in order to better verify the actual effect of our method, we carried out a case study. In Figure 2, we present a visualization of the results obtained from the

knowledge base retrieval process for attribute words, along with the model-predicted descriptions. In this visualization, we use the term "Truth" to represent the actual caption, "Predict" to present the generated caption of our model, and "KeyWords" to denote the attribute tags retrieved from the knowledge base.

The retrieved entity words from the knowledge base significantly impact the final description, contributing to the generation of more realistic descriptions. The generated content closely resembles the actual attributes present in the image. The blue portions in the visualization indicate the common parts between the model-generated description and the ground truth, highlighting the model's ability to generate more specific and contextually relevant descriptions that incorporate aspects such as categories and styles. It also includes additional relevant attributes without altering the original intent of the description, such as "heather grey", "leather lining", and "approx. 1.5" heel". This showcases the model's fine-grained capabilities.

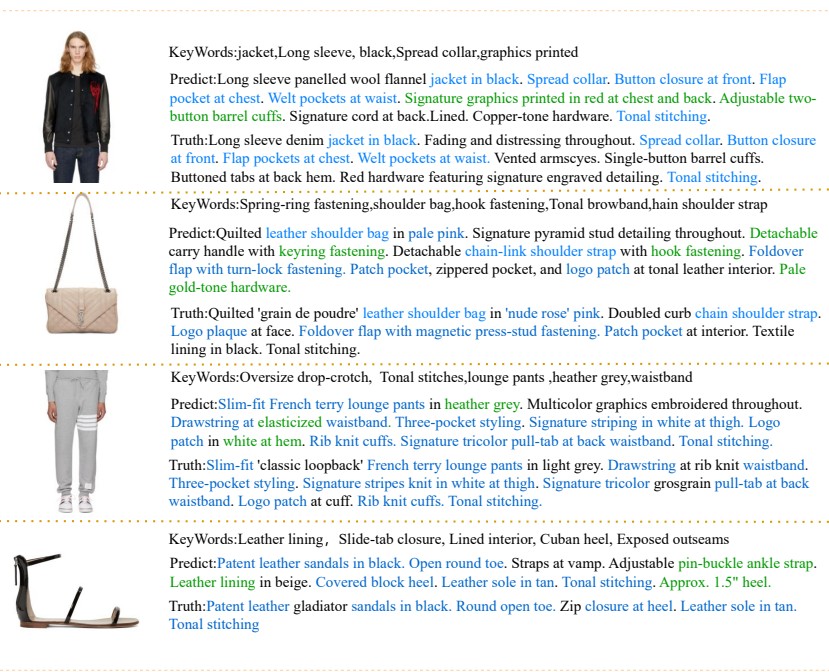

Figure 2: Comparison between the sentences predicted by the KEIC and the real description. The words in blue represent the same part as the true description, and the green part represents the more fine-grained representation words than the true description. These words are influenced by the external knowledge base.

## 5  CONCLUSIONS

Through this research, we develop a model called KEIC (i.e. knowledge enhanced image captioning) that can effectively learn the relationship between images and their corresponding captions and generate high-quality descriptions for fashion products. By incorporating additional knowledge into the image captioning task, KEIC enhances the model's ability to generate more accurate and informative captions. Through extensive experiments and evaluations, we demonstrate the effectiveness of KEIC in generating meaningful and contextually relevant captions for product images. The results highlight the advantages of incorporating domain knowledge and fine-grained features in the image captioning task, showcasing the potential of our approach to improve the performance and accuracy of image captioning systems. In future work, we aim to delve deeper into the task of fine-grained label extraction for product images, focusing on accurately identifying unique geographical indicators for different products in the e-commerce domain and generating corresponding descriptions.

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
