# OpenReview forum: "Knowledge Enhanced Image Captioning for Fashion Products"
_ICLR.cc/2025/Conference — Submitted to ICLR 2025_

### Official Review · Reviewer_USUu · 2024-10-20

**Soundness:** 2
**Presentation:** 2
**Contribution:** 1
**Rating:** 3
**Confidence:** 4

**Summary:**

1. This paper incorporate a knowledge base to a end-to-end architecture. The knowledge base includes attribute tags and their text features.
2. To extract relevant tags, the authors use two similarity scores based on relevance and diversity.
3. The authors design a new mode mapping network, which concatenates the attribute tags embedding with image features and passes them into a attention network.
4. During generation, the authors first use beam search, find the most similar candidate sentences based on CLIP score.

**Strengths:**

1. A nice work to apply image caption models to fashion product domain
2. Good combination of existing ideas

**Weaknesses:**

1. The comparative methods selected in the main results are somewhat dated, with the most recent work being ClipCap from 2021, which may not fully reflect the latest advancements in the field. Suggestion: It would be beneficial to include more recent caption model for comparison such as BLIP2, mPLUG, etc.
2. The contributions could be more clearly emphasized, and the methods employed show limited novelty. While useful, the techniques used may not stand out as major contributions to the field such as your filtering strategy.
3. The experimental design could be expanded. Main results: The current study uses only one dataset. Ablation study: It mainly focuses on the inclusion of knowledge-based methods and filtering strategies. Suggestions: Consider exploring a wider variety of large language models (as many open-source LLMs are available, LLama3, etc) and experimenting with different similarity scores.
4. The methods section might be slightly more detailed than necessary for the simplicity of the approach. It dedicates a significant portion of text and mathematical formulas to explain your methods. Suggestions: Instead of adding complexity to the methods, it may be more beneficial to allocate additional effort toward conducting more comprehensive experiments as suggested in (3).

**Questions:**

No questions.

---

### Official Review · Reviewer_iwfJ · 2024-11-02

**Soundness:** 2
**Presentation:** 2
**Contribution:** 2
**Rating:** 3
**Confidence:** 4

**Summary:**

This paper integrates a knowledge base into the widely adopted end-to-end architecture, aims to enhance the availability of comprehensive data about fashion items. This idea is reasonable.

However, many method details are missing, making the proposed method hard to follow. The experiments are not very convincing, as the compared methods are old.

**Strengths:**

This paper integrates a knowledge base into the widely adopted end-to-end architecture, aims to enhance the availability of comprehensive data about fashion items. This idea is reasonable.

**Weaknesses:**

1. The equations are not very professional. If the symbol comes from a word, it should not be italic. Please revise Eq.3 to be clearer, Eq.3 should contain clear calculations of the rel(I,t).  Please also check other equations accordingly.

2. The Section Method is hard to follow. The flow from input(Image) to the output is not clearly described. For example, How the commodity knowledge base is contructed? Is F_t one of the input to Self-Attention module? Why calculating the similarity between F_I and F_t in Stage 'Feature Encoder'? In Stage 'Mode Mapping Network', where is the text comes from?  Is it come from the commodity KB?
In Stage 'Inference and Generation', why not using more advanced GPT model?

3. In Table 1, no difinition and citation for GSA-GED. Other compared methods are old.

4. The main novelty is not well explained. For example, why and how the commodity knowledge base is contructed? Please provide more details about the proposed method.

**Questions:**

Pleae see the Weaknesses

---

> ### Comment · Reviewer_iwfJ · 2024-11-27
>
> No rebuttal material is provided. I will keep my decision to reject this submission considering its current draft.

---

### Official Review · Reviewer_2irx · 2024-11-03

**Soundness:** 3
**Presentation:** 3
**Contribution:** 1
**Rating:** 3
**Confidence:** 5

**Summary:**

+ This system would be very useful in e-commerce applications. Business owners often spend a lot of time writing descriptions of the products. This system would reduce such burdens.
+ The examples in Figure 2 are promising, as they capture or utilize concepts similar to those in the ground truth data. It would be beneficial if the authors could provide additional examples.

**Strengths:**

+ This system would be very useful in e-commerce applications. Business owners often spend a lot of time writing descriptions of the products. This system would reduce such burdens.
+ The examples in Figure 2 are promising, as they capture or utilize concepts similar to those in the ground truth data. It would be beneficial if the authors could provide additional examples.

**Weaknesses:**

- Technical novelty is limited.
  - While the overall framework could be valuable in e-commerce contexts, the paper lacks clear technical innovations, as it primarily combines simple embedding and distance calculations. The authors are encouraged to clarify their unique technical contributions more concretely, without which the proposed method could not be realized.

- The technical details require further clarification.
  - In Eq. (4), a diversity score is defined. However, this score only taking the maximum in Sim(Ft, Fs), which seems too simple. The authors might want to clarify why this approach was employed. It would also be helpful if the authors could consider comparing their method with other diversity-related measures, such as entropy or inverse document frequency (IDF).
  - There are many hyper-parameters in the proposed method such as $\lambda$, $k$, $p$, etc. In Section 4.1.3, their values are presented and explained that those values are decided by empirical study. The authors could strengthen their analysis by showing the sensitivity of these hyper-parameters or presenting performance comparisons based on different parameter values. In other words, it would be helpful if the authors could show a table or a graph showing how performance varies across different values for key hyperparameters like $\lambda$, $k$, and $p$.
  - Eq. (8) lacks clarity, particularly in the concatenation of $F_I$ and $F_S$. Since $F_I$ is an $l \times 1$-dimensional feature from Eq. (1) and $F_S$ is a set of features from Eq. (7), their concatenation is not straightforward. Additionally, the parameters and training procedure for the MLP function need further explanation.
  - The explanations in lines 252-255 were not clear. Could the authors provide more details on how the mapping network is trained?
  - The GPT-2 language model is used in the final stage of the proposed method. However, it is natural to question why more recent GPT models or other language models were not considered. Providing reasons for this choice or showing comparative results would strengthen the paper.

- The experimental comparison is insufficient and could be strengthened with further analysis.
  - I am a bit confused about the initial tag set. Judging from Eq. (1), I understood that the initial tag set is extracted by using a CILP model. I wonder how the CLIP model can generate such diverse keywords as shown in Figure 2. If the tag sets come from Eq. (6), I would like the authors to discuss the accuracy of the extracted keywords. In addition, if the extracted keywords are not related to the products, the generated descriptions would also be wrong. The authors might want to discuss such erroneous cases by showing the percentage of such cases or by showing the failure cases. In Figure 2, only successful cases are shown.
  - Related to the above, tag extraction uses a simple maximum search in Eq. (6) and a $k$ nearest neighbor search. There is no guarantee that the extracted tags are accurate or relevant to the image. For instance, the tag “red” might appear for an apple image, while the input image might actually be a green apple. The accuracy of this tag retrieval and the impact of erroneous tags on the generated captions should be discussed.
  - It would be helpful if the authors provided examples of the top-$k$ nearest neighbor search results, as no concrete examples are currently shown.
  - The baseline methods are relatively outdated (2015-2021), and the study could benefit from including more recent methods. At a minimum, comparisons with GPT-3.5, GPT-4, and LLaVA should be added.
  - The improvements over baseline methods are minimal, as seen in Table 1, particularly with GSA-GED and ClipCap, where most gains are within 1 point. The authors might want to provide specific examples of captions where their method outperforms the baselines.


Here are some comments to improve the paper:
- The text in the figures is too small and should match the font size used in the main body. Please adjust the figure text to improve readability without requiring magnification.
- There is an extraneous “]” in Eq. (4) that should be removed.
- The Fashion-Gen dataset reference is missing and should be added to ensure proper citation.
- There are inconsistencies in the reference styles, such as varying capitalization for “International Conference on Machine Learning” and the selective inclusion of the publisher name (PMLR) for ICML. Please standardize the reference formatting across all citations.

**Questions:**

Please answer my comments and questions in the Weakness part.

---

> ### Comment · Reviewer_2irx · 2024-11-27
>
> So far, no rebuttal from the authors has been provided. I will keep my decision to reject.

---

### Official Review · Reviewer_Mpfy · 2024-11-03

**Soundness:** 2
**Presentation:** 3
**Contribution:** 2
**Rating:** 3
**Confidence:** 5

**Summary:**

This paper proposes a new method for fashion image captioning, integrating a knowledge base into an end-to-end architecture to combine attribute and image features. A filtering strategy further refines descriptions by selecting the best candidate sentences, with experiments showing superior performance over existing methods.

**Strengths:**

1. The authors integrate a rich fashion knowledge base into the end-to-end architecture, enhancing understanding with multidimensional data.
2. The authors design a mode mapping network that fuses attribute features from the knowledge base with image features, achieving a comprehensive representation of fashion items.
3. The authors introduce a filtering strategy to improve description quality by selecting the best candidate sentence from beam search results.
4. Their experimental results, including comparisons, ablation studies, and case analyses, confirm the effectiveness of the proposed method.

**Weaknesses:**

1. The major contribution could be the integration of a knowledge base and the mode mapping network. However, the technical novelty is relatively weak in the proposed design. The use of external knowledge base and similar feature fusion is commonly used in vision language tasks. Have you considered what unique design could be involved, which is tailored for fashion domain?

2. The task itself is a little bit limited. Recent fashion VLMs normally handle a batch of Fashion tasks, including fashion captioning, I2T/T2I retrieval, TGIR, etc. Focusing on fashion image captioning itself is limited.

3. The current evaluation is limited, for instance, more datasets can be considered.

Overall, I currently believe the contribution of this submission is limited, therefore the current version is not a valid ICLR publication.

**Questions:**

Please see Weaknesses

---

### Meta-Review · Area_Chair_GitQ · 2024-12-18

**Metareview:**

This paper studies the problem of fashion image captioning by integrating a knowledge base into an end to end architecture. The reviewers appreciate the design of a fashion knowledge base integration, which creates a compreshensive representation of fashion items. The design filtering strategy enhances description quality and experimental results confirm its effectiveness. This method is particularly useful for e-commerce business owners.

Despite all this, several questions and concerns are raised. 1. Technical novelty and clarity issues. The paper lack clear technical innovations and relies on the commonly used methods in VL tasks. 2. Evaluation and comparsion limitations. The evaluation is limited to a signal task and dataset, and comparison are made with outdated baselines. 3. Presentation problems. Given all these, I think this paper is not ready for publication.

**Additional Comments On Reviewer Discussion:**

No rebuttal provided.

---

### Decision · Program_Chairs · 2025-01-22

Reject